# Unraveling Genetic Variation and Inheritance Patterns in Newly Developed Maize Hybrids for Improving Late Wilt Disease Resistance and Agronomic Performance Under Artificial Inoculation Conditions

**DOI:** 10.3390/life14121609

**Published:** 2024-12-05

**Authors:** Nasr A. Ghazy, Mayasar I. Al-Zaban, Fatmah Ahmed Safhi, Maha Aljabri, Doaa A. Kafsheer, Imen Ben Abdelmalek, Mohamed M. Kamara, Elsayed Mansour, Salem Hamden

**Affiliations:** 1Maize and Sugar Crops Diseases Department, Plant Pathology Research Institute, Agricultural Research Center, Giza 12112, Egypt; nasrghazy@yahoo.com (N.A.G.); doaa74147@gmail.com (D.A.K.); 2Department of Biology, College of Science, Princess Nourah bint Abdulrahman University, Riyadh 11671, Saudi Arabia; mialzaban@pnu.edu.sa (M.I.A.-Z.); faalsafhi@pnu.edu.sa (F.A.S.); 3Department of Biology, Faculty of Science, Umm Al-Qura University, Makkah 24231, Saudi Arabia; myjabri@uqu.edu.sa; 4Department of Biology, College of Science, Qassim University, Buraydah 51452, Saudi Arabia; 5Department of Agronomy, Faculty of Agriculture, Kafrelsheikh University, Kafr El-Sheikh 33516, Egypt; mohamed.kamara@agr.kfs.edu.eg; 6Department of Crop Science, Faculty of Agriculture, Zagazig University, Zagazig 44519, Egypt; 7Agricultural Botany Department (Plant Pathology), Faculty of Agriculture, Kafrelsheikh University, Kafr El-Sheikh 33516, Egypt

**Keywords:** disease resistance, agronomic traits, maize breeding, line×tester analysis, hybrid classification, genetic parameter estimates, sustainability

## Abstract

Late wilt disease caused by the fungal pathogen *Magnaporthiopsis maydis* represents a major threat to maize cultivation in the Mediterranean region. Developing resistant hybrids and high-yielding offers a cost-effective and environmentally sustainable solution to mitigate yield losses. Therefore, this study evaluated genetic variation, combining abilities, and inheritance patterns in newly developed twenty-seven maize hybrids for grain yield and resistance to late wilt disease under artificial inoculation across two growing seasons. The results indicated highly significant variations among assessed hybrids for all measured traits. Combining ability analysis identified IL-306, IL-304, and IL-303 as excellent combiners for grain yield and late wilt resistance, positioning them as superior candidates for hybrid development. Additionally, IL-302 was identified as a strong general combiner for earliness, and IL-307 and IL-309 demonstrated potential for producing short-statured hybrids critical for improving lodging tolerance and maximizing yield. Specific combining ability effects indicated promising earliness, yield, and disease-resistance hybrids, including IL-303×T2 and IL-306×T1. GGE biplots presented optimal line×tester combinations, offering strategic guidance for hybrid development. The principal component analysis demonstrated strong associations between grain yield, late wilt resistance, and key agronomic traits, such as ear length and kernel number. The observed robust positive association between grain yield, late wilt resistance, and yield attributes suggests selection potential for improving maize productivity. Moreover, the genotypic correlations revealed that earlier silking, taller plants, and higher kernel counts were strongly linked to enhanced yield potential. Genetic parameter estimates indicated a predominance of non-additive genetic effects for most traits, with moderate to high broad-sense heritability suggesting substantial genetic contributions to phenotypic variance. This research provides valuable insights to support the development of disease-resistant and high-yielding maize hybrids addressing critical food security challenges.

## 1. Introduction

Maize (*Zea mays* L.) is one of the essential cereal crops globally, playing a vital role in feed, food, and fuel production systems [1]. Over time, it has transitioned from being a staple in subsistence farming to a driving force in local and national economies, highlighting its essential contribution to agriculture and economic development worldwide [2]. However, the ongoing changes in global climate are expected to exacerbate various biotic stresses, which can severely limit maize productivity [3].

Among the diseases threatening maize production, late wilt is one of the most devastating [4]. This disease, caused by the soilborne fungus *Magnaporthiopsis maydis*, poses a severe challenge to maize cultivation due to its impact on plant vascular systems, leading to wilting and subsequent plant death [5]. Late wilt was first identified in Egypt, which continues to be a major concern for maize farmers [6]. The disease increases under specific environmental conditions, particularly in dry regions where maize is widely grown. These include areas like India, Egypt, and the Iberian Peninsula, which are increasingly susceptible to late wilt infections as rising temperatures, driven by climate change, create more favorable conditions for its proliferation [7]. In Egypt, the impact of late wilt has been extensively studied, with findings showing a strong positive correlation between grain yield losses and late wilt incidence [8]. Yield reductions in infested fields can reach up to 40%, posing a significant threat to food security and agricultural sustainability [9]. Furthermore, the economic implications of late wilt are substantial, as it reduces yields and necessitates increased expenditure on disease management practices. This emphasizes the urgency of developing effective and sustainable strategies to mitigate its impact on maize production globally.

Climate change has significantly influenced the prevalence and severity of plant diseases, including late wilt disease in maize [10]. Rising global temperatures, altered precipitation patterns, and increasing water stress associated with climate change create conditions that favor the pathogen’s survival and proliferation [11]. High soil temperatures and prolonged drought periods are becoming more common due to global warming, exacerbating the disease by weakening maize plants and making them more susceptible to infection [12]. Mediterranean regions are already facing water scarcity and have reported an alarming increase in the incidence of late wilt disease, threatening food security in these vulnerable areas [13]. Additionally, climate-induced changes in planting schedules and cropping systems can further enhance disease spread by extending the pathogen survival window and altering its epidemiology [14]. Hence, late wilt disease remains a major threat under these changing climatic conditions. The fungus *Magnaporthiopsis maydis* induces rapid wilting in maize, typically beginning prior to tasseling and continuing until near maturity [15]. Symptoms of late wilt disease include premature leaf wilting, stalk discoloration, and, in advanced stages, stalk tissue disintegration and fibrous breakdown [16]. Initial signs of the disease appear around 60 days after planting and persist until the plant nears maturity. The infection starts on the lower leaves and gradually moves upward, causing the tissues between the veins to turn pale green, followed by inward curling of the leaves [17]. As the disease progresses, yellow to reddish-brown streaks develop on basal internodes of the stalk, causing shrinkage and desiccation. Internally, brown discoloration appears along the internodes, further weakening the plant. Late wilt significantly reduces ear formation; any ears that develop are often deformed or poorly filled. The pathogen can severely infect the kernels, leading to seed rot and damping off during early plant stages [18]. While fungicide application provides partial control, its use often leaves harmful pesticide residues, disrupts beneficial soil microbes, and poses environmental and human health risks [19]. As a result, breeding resistant maize hybrids is considered a more sustainable and cost-effective solution for mitigating yield losses caused by late wilt disease.

Most hybrids resistant to late wilt disease tend to have lower yields or undesirable agronomic traits, making it crucial to find hybrid combinations that achieve resistance to late wilt and high yields [20]. Maize breeders develop new hybrids using elite inbred lines and testing them to identify superior hybrids suited to different conditions. The performance of parent lines and their hybrids is key for estimating combining ability, assessing hybrid potential, and selecting superior parents for hybrid development [21,22,23]. Identifying optimal parental combinations is essential for breeding hybrids resistant to late wilt disease. The line×tester mating design is a powerful tool for assessing the effects of general combining ability (GCA) and specific combining ability (SCA), helping breeders identify suitable parental lines and understand the genetic mechanisms governing key traits, even with limited sample sizes [24,25]. This approach facilitates the selection of excellent parents for producing high-yielding, disease-resistant hybrids. By analyzing GCA and SCA variances, researchers can differentiate the roles of additive and non-additive gene effects in the expression of target traits [26,27]. GCA reflects additive gene effects, while SCA indicates non-additive effects, which influence hybrid performance relative to their parent lines. Research suggests that additive genetic effects play a significant role in late wilt disease resistance [28], while non-additive gene action holds greater significance in determining grain yield under various environmental conditions [29,30,31]. However, there is still a lack of detailed understanding regarding the interaction between late wilt resistance and high yield potential in maize hybrids. This study aimed to (1) evaluate the GCA of selected lines and testers and the SCA of their hybrids under artificial inoculation conditions; (2) investigate the gene action involved in yield traits and late wilt disease resistance; and (3) identify high-yielding hybrids that exhibit strong resistance to late wilt disease.

## 2. Materials and Methods

### 2.1. Plant Material and Generation of F_1_ Hybrids

Nine genetically diverse maize (*Zea mays* L.) lines were sourced from CIMMYT and the Agricultural Research Center in Egypt (Appendix A). During the summer of 2021, these nine inbreds were hybridized with three high-yielding commercial hybrids: SC-167 (T1), Pioneer-3062 (T2), and TWC-360 (T3), employing a line×tester mating design, resulting in the production of 27 F1 maize hybrids.

### 2.2. Inoculum Preparation of Magnaporthiopsis maydis

Isolates of *M. maydis* were collected from symptomatic maize plants in different locations across Egypt, where late wilt disease was observed. The isolates were confirmed as *M. maydis* based on cultural and microscopic characteristics, with pathogenicity verified following the method of Samra et al. [5]. For inoculum preparation, 150 g of sorghum seeds were soaked overnight in water within 500 mL glass bottles, after which the excess water was drained, and the seeds were autoclaved for one hour. A mycelial disc from *M. maydis* culture grown on potato dextrose agar (PDA) with 0.2% yeast extract was introduced into each bottle. The cultures were incubated at 27 °C for seven days [32], which allowed the fungus to grow for 15 days. The contents were then thoroughly mixed to ensure a homogenized inoculum, which was subsequently utilized for soil infestation following the procedure of Sabet et al. [33].

### 2.3. Field Evaluation Under Artificial Inoculation

The 27 F1 crosses and check hybrid TWC-368 were assessed artificially inoculated with *Magnaporthiopsis maydis* in a disease nursery. The nursery was inoculated each year with different clonal lineages of *M. maydis* following the methodology outlined by Zeller et al. [15] to enhance selection efficiency in differentiating between susceptible and resistant hybrids. The trials were performed during the 2022 and 2023 growing seasons at Sakha Agricultural Research Station, Egypt (30°3′ N, 31°3′ E). The meteorological conditions for crossing and evaluation seasons are detailed in Figure 1.

Randomized Complete Block Design in three replicates was applied in both years. Each plot consisted of two rows, each 6 m long, with row spacing of 0.70 m and plant spacing of 0.25 m within rows. Twenty-one days after planting, seedlings were thinned to a single plant per hill to achieve optimal plant density. Standard furrow irrigation was used, following regional agricultural practices. Phosphorus was applied as calcium superphosphate (15.5% P_2_O_5_) at 75 kg P_2_O_5_/ha during seedbed preparation, and potassium sulfate (48% K_2_O) was applied at 115 kg K_2_O/ha after thinning. Nitrogen was supplied as urea and split into equal applications during the first and second irrigations.

### 2.4. Data Collection

#### 2.4.1. Agronomic Traits

Days to silking were determined when 50% of the plants in each plot had emerged silks. Plant height (cm) was recorded from ground level to tassel top, and ear height (cm) was recorded from plant base to node-bearing primary ear. At harvest, ten ears were randomly selected per plot for further agronomic evaluation, including ear length, number of kernels/row, and rows/ear. Ear length (cm) was determined from base to tip after husk removal. The number of kernels/row and rows/ear were counted and averaged. Plots were harvested manually, and grain yield (t/ha) was determined based on the weight of shelled grain, standardized to a moisture content of 15.5%.

#### 2.4.2. Late Wilt Disease Resistance

Late wilt disease symptoms were monitored 35 days after 50% silking, following the identification method of El-Shafey et al. [18]. The number of infected plants per plot was recorded, and the percentage of resistance to late wilt was calculated using the formula:$$\begin{array}{cc} lat\mathrm{e-}wilt\;resistance\;(\%) & \\ & =(\frac{Number\;of\;uninfected\;plants\;in\;each\;plot}{Number\;of\;total\;plants\;in\;each\;plot})\times 100 \end{array}$$

Resistance percentages were classified into five categories: 100–90% (resistant), >90–80% (moderately resistant), >80–70% (moderately susceptible), >70–50% (susceptible), and >50–0% (highly susceptible).

### 2.5. Statistical Analysis

Analysis of variance (ANOVA) was performed using SAS software (version 9.1). Line×tester analysis, based on Kempthorne [34], was utilized to estimate the GCA of lines and testers and SCA of hybrids. Standard heterosis was calculated by comparing F1 performance to the check hybrid using the formula: Heterosis over the standard check = [(F_1_ − SC)/SC] × 100. Principal component and cluster analyses were generated utilizing R software (version 4.2.2).

## 3. Results

### 3.1. Variance Analysis

Table 1 presents mean squares derived from separate and combined analyses of variance for studied characters of assessed maize hybrids under artificial inoculation conditions across two seasons. The analysis revealed highly significant differences in mean squares between the two years for all traits except for the number of rows/ear (Table 1). Significant differences were observed among the assessed hybrids across all traits (Table 1). The interaction between year and crosses was significant for all characters except the number of kernels/row and ear length. Furthermore, significant effects were revealed for lines, testers, and their interaction across all characters. Additionally, the interactions involving lines×years, testers×years, and line×tester×year were significant for most measured traits, excluding number of kernels per row and ear length.

### 3.2. Mean Performance

Figure 2 and Figure 3 illustrate the comparative performance of maize hybrids under artificial inoculation conditions with *M. maydis*. The figures emphasize the variability among hybrids in their performance across studied traits, displaying their differential adaptability under the tested conditions. Days to silking exhibited noticeable variability across the two years, averaging 64.26 days in the first year (ranging from 60.33 to 69.00 days), 62.69 days in the second year (ranging from 58.67 to 68.00 days), and 63.48 days across both years (ranging from 59.83 to 67.17 days). In the first year, the earliest silking hybrid was IL-308×T1, followed by IL-302×T3, IL-302×T1, and IL-303×T1, while the latest silking hybrid was IL-309×T2, IL-308×T2, IL-303×T3, and IL-304×T3 (Figure 2A). In the second year, the earliest silking hybrids were IL-303×T1, IL-302×T1, IL-302×T2, and IL-308×T2, whereas the latest silking hybrids were IL-308×T3, IL-309×T3, IL-306×T3, and IL-303×T3. Plant height showed considerable variability across the two years, with an average of 259.23 cm in the first year (varying from 220.00 to 283.33 cm), 254.94 cm in the second year (varying from 210.00 to 288.33 cm), and 257.08 cm across both years (varying from 215.00 to 285.83 cm). In the first year, the shortest hybrids were IL-309×T3, 303×T3, IL-307×T2, and IL-307×T3, although the tallest hybrids were IL-304×T3, IL-304×T1, IL-306×T1, and IL-302×T1 (Figure 2B). In the second year, the shortest hybrids were IL-309×T3, 307×T2, IL-302×T2, and IL-303×T1, while the tallest hybrids were IL-304×T3, IL-304×T1, IL-306×T1, and IL-309×T2. Ear height showed variation across the two years, with an average of 151.19 cm in the first year (fluctuating from 116.67 to 190.00 cm), 146.85 cm in the second year (fluctuating from 115.00 to 188.33 cm), and 148.57 cm across both years (fluctuating from 115.83 to 189.17 cm). In the first year, the lowest ear position was observed in IL-309×T3, IL-302×T3, IL-301×T3, and IL-305×T3, whereas the highest ear position was recorded in IL-304×T1, IL-306×T3, IL-302×T1, and IL-306×T1 (Figure 2C). In the second year, the lowest ear height was observed in IL-309×T3, IL-302×T3, IL-301×T2, and IL-309×T2, while the highest ear height was recorded in IL-304×T1, IL-302×T2, IL-308×T2, and IL-306×T1. Ear length exhibited notable variability across the two years, with an average of 17.83 cm in the first year (differed from 13.67 to 22.26 cm), 16.54 cm in the second year (differed from 13.50 to 20.99 cm), and across both years, 17.19 cm (differed from 14.33 to 20.58 cm). The hybrids with the longest ears in the first year were IL-306×T1, IL-308×T1, IL-304×T1, and IL-309×T1, demonstrating superior performance in this trait, whereas the hybrids with the shortest ears were IL-304×T3, IL-301×T3, IL-305×T2, and IL-304×T2 (Figure 2D). In the second year, the longest ears were recorded for IL-305×T1, IL-302×T1, IL-307×T1, and IL-304×T1; though, the hybrids with the shortest ears were IL-309×T3, IL-305×T2, IL-307×T3, and IL-304×T2.

The number of rows per ear displayed considerable variability across the two years, with an average in the first year of 15.31 (varying from 12.00 to 17.00), 14.82 in the second year (varying from 11.00 to 18.00), and 15.07 across both years (varying from 12.00 to 17.00). In the first year, the hybrids IL-306×T2, IL-305×T1, IL-303×T2, and IL-302×T2 had the highest number of rows per ear, whereas IL-309×T3, IL-308×T3, IL-305×T2, and IL-307×T3 recorded the lowest values (Figure 3A). Similarly, in the second year, the hybrids IL-301×T2, IL-306×T1, IL-303×T1, and IL-302×T2 achieved the highest number of rows, while IL-307×T3, IL-301×T3, IL-303×T3, and IL-305×T2 recorded the lowest. The average number of kernels per row was 42.49 in the first year (ranging from 34.00 to 46.67), 40.36 in the second year (ranging from 32.33 to 45.67), and 41.43 across both years (ranging from 34.17 to 46.17). In the first year, the hybrids IL-302×T2, IL-303×T2, IL-306×T1, and IL-304×T3 produced the highest number of kernels per row, while IL-307×T3, IL-307×T2, IL-301×T3, and IL-309×T3 recorded the lowest counts (Figure 3B). Similarly, in the second year, IL-303×T2, IL-306×T1, IL-304×T1, and IL-308×T2 exhibited the highest kernel counts, whereas IL-307×T2, IL-301×T3, IL-307×T3, and IL-308×T3 had the lowest values. Grain yield exhibited considerable variations across the two years, with an average of 10.32 t/ha in the first year (differing from 7.35 to 13.25 t/ha), 9.77 t/ha in the second year (differing from 7.17 to 12.13 tons/ha), and 10.10 tons/ha across both years (differing from 7.41 to 12.69 tons/ha). In the first year, the hybrids IL-306×T1, IL-303×T2, IL-306×T3, and IL-304×T1 achieved the highest grain yields, whereas IL-308×T2, IL-309×T3, IL-307×T2, and IL-305×T2 recorded the lowest yields (Figure 3C). Similarly, in the second year, the hybrids IL-303×T2, IL-306×T1, IL-301×T1, and IL-304×T2 exhibited the highest yields, while IL-309×T3, IL-303×T3, IL-309×T2, and IL-308×T3 had the minimal values. Late wilt disease resistance averaged 95.72% in the first year (fluctuating from 80.50% to 100%), 95.25% in the second year (fluctuating from 87.20% to 100%), and 95.51% across both years (fluctuating from 87.00% to 100%). In the first year, the most resistant hybrids included IL302×T2, IL303×T2, IL304×T1, IL305×T1, IL306×T1, IL306×T2, IL307×T2, IL308×T1, IL309×T1, and IL309×T2 (Figure 3D). In the second year, the top-performing hybrids were IL301×T1, IL303×T2, IL304×T1, IL304×T2, IL306×T1, and IL308×T2.

### 3.3. Hybrid Classification Based on Yield Characters and Late Wilt Disease Resistance

The assessed maize hybrids were classified based on yield traits into three distinct clusters, as illustrated in Figure 4a. Group A consisted of 9 hybrids (IL-303×T2, IL-306×T1, IL-304×T1, IL-302×T2, IL-301×T1, IL-304×T3, IL-306×T3, IL-304×T2, and TWC-368), which exhibited superior yield and contributing traits. Group B comprised 12 hybrids (IL-303×T1, IL-307×T1, IL-301×T2, IL-303×T3, IL-306×T2, IL-302×T3, IL-309×T1, IL-307×T3, IL-305×T3, IL-302×T1, IL-308×T3, and IL-305×T1) characterized by an intermediate yielding trait. Group C comprised 7 hybrids (IL-309×T3, IL-308×T2, IL-309×T2, IL-305×T2, IL-308×T1, IL-301×T3, and IL-307×T2) with the lowest yielding traits. Likewise, the hybrids were grouped into three clusters based on resistance to late wilt disease (Figure 4b). Group A contained 14 hybrids (IL303×T2, IL304×T1, IL306×T1, IL307×T1, IL301×T1, IL306×T2, IL309×T1, IL309×T2, IL307×T2, IL307×T3, IL305×T2, IL308×T2, IL305×T1, and IL304×T2), which exhibited superior resistance to late wilt disease. Group B comprised 8 hybrids (TWC-368, IL-308×T3, IL-308×T1, IL-304×T3, IL-302×T1, IL-306×T3, IL-302×T2, and IL-303×T3) depicted by high resistance to late wilt disease. Group C comprised 6 hybrids (IL-302×T3, IL-301×T3, IL-309×T3, IL-301×T2, IL-303×T1, and IL-305×T3) that showed the lowest resistance to late wilt disease.

### 3.4. GCA Effects

Positive GCA effects are desirable for most assessed characters, except for plant height, days to silking, and ear height, where negative values are preferred. No parent simultaneously demonstrated ideal GCA effects for all traits (Table 2). Line IL302 consistently showed significant and desirable negative GCA effects for days to silking across both years. Lines IL307 and IL309 exhibited the most significant negative GCA effects for plant height and ear height over the two years. Lines IL306 and IL308 displayed the most significant positive GCA effects across both years for ear length. In terms of number of rows/ear, the IL301, IL303, and IL306 displayed significant and desirable positive GCA. The inbreds IL302, IL303, IL304, and IL306 showed the most significant positive GCA effects across both years for the number of kernels/row, and IL303, IL304, and IL306 for grain yield. The lines IL304, IL306, and IL307 were identified as the top combiners for late wilt disease resistance across both years. Among the testers, T1 (SC-167) demonstrated the most significant and favorable GCA effects for the number of kernels/row, days to silking, and grain yield. T2 (Pioneer-3062) showed the highest GCA effects for a number of rows/ear. T3 (TWC-360) excelled in GCA effects for plant and ear height. T1 and T2 consistently demonstrated the highest significant and desirable GCA effects for late wilt disease resistance across both years.

### 3.5. SCA Effects

Maize hybrids exhibited varying SCA effects across all measured characters (Table 3 and Table 4). The inbreds IL302×T3, IL303×T2, and IL306×T3 displayed significant negative SCA effects for days to silking during the first year, while IL304×T3 and IL309×T2 in the second year and IL303×T1, IL304×T2, IL305×T3, and IL308×T1 across both years (Table 3). These hybrids show promise for maize breeding programs targeting earliness. The highest desirable negative SCA effects for plant height were noted in hybrids such as IL302×T2, IL307×T2, IL308×T1, and IL309×T3 across both years. The most favorable negative SCA effects for ear height were detected in hybrids IL302×T3, IL304×T2, and IL306×T1 across both years. Hybrids IL304×T1 and IL309×T2 demonstrated consistently significant positive SCA effects for ear length across both years. Hybrids IL304×T3 and IL305×T1 demonstrated significantly high SCA effects for the number of rows/ear (Table 4). IL304×T3 and IL307×T1 displayed highly significant positive SCA effects in the first year, while IL301×T1 and IL305×T3 in the second year. The hybrids IL302×T2, IL308×T3, and IL309×T2 exhibited the most favorable SCA effects for grain yield in the first season. In the second season, IL301×T1 and IL305×T3 demonstrated remarkable SCA effects, with IL303×T2 and IL306×T1 consistently showing strong positive effects across both years. IL302×T2, IL303×T2, IL304×T3, and IL305×T1 exhibited the highest significant and positive SCA effects for late wilt disease resistance in the first year. While in the second season, IL301×T1 displayed the highest resistance, while IL301×T1 and IL303×T2 consistently demonstrated strong resistance across both years.

### 3.6. Best Hybrids Between Lines and Testers Using GGE

GGE biplots explained a substantial proportion of the variation in the evaluated traits of assessed hybrids, with explained variation ranging from 77.42% for ear height to 90.61% for plant height across the two years (Figure 5 and Figure 6). The polygon view of the GGE biplot effectively illustrated interaction patterns between lines and testers, enabling the identification of suitable mating partners and the highest-performing hybrids. Inbred lines located at the vertex of a sector were identified as the best match for the testers within that sector. In contrast, lines near the origin of the biplot exhibited low responsiveness to testers, indicating weaker performance. Vertex inbred lines situated in sectors without testers reflected limited compatibility, demonstrating their minimal performance with the available testers. The biplot for days to silking categorized the hybrids into five sectors, with IL-305, IL-307, IL-308, IL-303, and IL-302 positioned at the vertices (Figure 5A). Notably, no testers were present in the sectors of IL-305 and IL-302, indicating these inbred lines were not suitable mating partners for the available testers. The highest performance could be observed for IL-307 when crossed with T1 and T2 and for IL-308 and IL-303 when crossed with T2. For plant height, the biplot identified IL-304, IL-302, IL-307, IL-309, and IL-301 as vertex entries, with no testers located in the sectors of IL-302, IL-307, or IL-309, suggesting these lines were minimal partners (Figure 5B). The highest performance was observed for IL-304 with T3 and IL-306 with T2. Similarly, in the biplot for ear height, the vertex entries included IL-302, IL-306, IL-303, IL-307, and IL-309, but only IL-302 (crossed with T1 and T2) and IL-306 (crossed with T3) exhibited the highest performance, indicating optimal mating compatibility (Figure 5C). For ear length, IL-305 was the vertex inbred for T1 and T3, while IL-302 was the vertex for T2 (Figure 5D). Regarding the number of rows/ear, the highest performance was assigned to IL-306 crossed with both T1 and T3 and IL-301 crossed with T2 (Figure 6A). Similarly, for the number of kernels per row, IL-302, IL-306, and IL-304 with T1 displayed the highest performance (Figure 6B). In the case of grain yield, the vertex entries were IL-306, IL-304, IL-303, IL-309, and IL-308, with IL-306 showing the highest values when crossed with T1 and T3 and IL-303 with T2 (Figure 6C). For resistance to late wilt disease, the biplot revealed IL-307, IL-303, IL-302, and IL-301 as the vertex entries (Figure 6D). The highest performance was observed for IL-307 with T3, IL-303 with T2, and IL-301 with T1. The results from the GGE biplot analysis were consistent with the conventional line×tester analysis by Kempthorne, demonstrating its effectiveness in identifying superior hybrids and optimal line×tester combinations for breeding programs.

### 3.7. Standard Heterosis

The standard heterosis identified hybrids with significant effects in desirable directions across various traits. The considerable significant negative heterotic effects for days to silking, favoring earliness, were noted in IL302×T1, IL302×T2, IL302×T3, IL303×T1, IL304×T2, and IL308×T1 (Figure 7). Favorable heterotic effects for shorter plant stature were noted in IL302×T2, IL307×T2, and IL309×T3. Hybrids IL302×T3, IL305×T3, and IL309×T3 exhibited desirable negative heterosis, indicating lower ear placement. For ear length, IL302×T1, IL304×T1, IL305×T1, IL306×T1, IL306×T2, and IL308×T1 showed significant positive heterotic effects. IL301×T1, IL301×T2, IL302×T2, IL303×T1, IL303×T2, IL306×T1, and IL306×T2 presented significant positive heterotic effects for number of rows/ear. IL303×T2 and IL306×T1 demonstrated notable positive heterosis for the number of kernels/row. Hybrids IL303×T2 and IL306×T1 were identified as the most effective heterotic combinations for grain yield. The hybrids IL303×T2, IL304×T1, and IL306×T1 exhibited superior significant positive heterotic effects for late wilt disease resistance.

### 3.8. Genetic Parameter Estimates

The genetic variance components for studied characters are illustrated in Table 5. SCA variance was greater than GCA variance for most characters, except for ear height, days to silking, and number of rows/ear. This suggests that non-additive genetic effects such as dominance or epistasis are predominant in their inheritance. Additionally, dominance genetic variance was greater than additive genetic variance for these traits, further emphasizing the influence of non-additive gene action in their expression. Broad-sense heritability was moderately high for most traits, indicating a substantial genetic contribution to phenotypic variance. Meanwhile, narrow-sense heritability was consistently lower than broad-sense heritability, reflecting the limited contribution of additive genetic effects to total genetic variation for most traits. Narrow-sense heritability values for ear height, days to silking, and number of rows/ear were relatively high, suggesting a significant additive genetic component for these traits. However, narrow-sense heritability was minimal for other characters, indicating that environmental factors or non-additive genetic influences play a more prominent role in their expression.

### 3.9. Association Among Hybrids and Agronomic Characters

Principal component (PC) analysis explored the association between maize hybrids and evaluated characters. The first two PCs captured a significant proportion of total variance, with the first component (PC1) accounting for 60.10% and the second (PC2) accounting for 16.44%, as shown in the PCA biplot (Figure 8). The first principal component exhibited the greatest variation, clearly differentiating hybrids with negative and positive values. The hybrids positioned on the PC1 positive side, such as IL-304×T1, IL-306×T1, IL-302×T1, IL-301×T1, IL-303×T2, and IL-302×T2, were associated with superior yield traits under artificial inoculation with *M. maydis*. In contrast, hybrids located on the negative side, including IL-309×T3, IL-301×T3, and IL-307×T3, exhibited inferior yield traits. The close proximity of the vectors in the PC biplot depicted a robust positive relationship between grain yield and several key traits, principally the number of rows/ear, number of kernels/row, plant height, ear height, ear length, and resistance to late wilt disease. Conversely, there was a negative association with days to silking.

Moreover, genotypic correlation among studied traits revealed various significant relationships (Figure 9). A negative correlation between days to silking and key yield traits, including the number of kernels/row, number of rows/ear, and grain yield, suggests that earlier silking contributes positively to yield-related traits. Plant height demonstrated a significant positive relationship with the number of kernels/row. However, plant height was negatively associated with days to silking, suggesting taller plants are linked to earlier maturity. Grain yield exhibited strong positive correlations with several important characters, such as the number of rows/ear, number of kernels/row, and plant height. These relationships highlight the significance of these characters as contributors to yield potential. Grain yield also demonstrated a positive relationship with late wilt disease resistance, revealing that hybrids with higher resistance to the disease tend to produce better yields.

## 4. Discussion

Late wilt disease is among the most devastating threats to maize production in the Mediterranean region, with its impact expected to escalate as climate change progresses [10]. Climate change plays a pivotal role in shaping the development and spread of fungal diseases worldwide [35]. Rising temperatures, increased humidity, and shifting precipitation patterns create optimal conditions for the proliferation and geographic expansion of fungal pathogens [35,36]. For instance, higher temperatures can accelerate fungal life cycles, leading to more frequent and severe outbreaks. Similarly, changes in rainfall patterns can increase moisture levels, which are critical for spore germination and subsequent infection [37]. Hence, developing high-yielding maize hybrids with enhanced resistance to this disease offers a sustainable and environmentally friendly solution to mitigate yield losses and ensure food security. Exploring genetic variability, combining ability, and gene action is essential for breeding disease-resistant hybrids with improved yield traits. This study revealed significant variations among assessed maize hybrids in agronomic traits, yield performance, and late wilt disease resistance. This highlights substantial genetic diversity facilitating the selection of promising inbred lines and hybrids for breeding programs. Previous reports, such as Kamweru et al. [2], Bhatla et al. [38], Aboderin et al. [39], Kamara et al. [20], and Biradar [40], similarly documented significant genetic variability among maize hybrids in yield traits and late wilt disease resistance. The studied traits displayed significant interaction between assessed hybrids and growing seasons, indicating complex genetic control influenced by non-additive genetic effects. The PCA and GGE biplot analyses further confirmed these interactions by effectively differentiating hybrids based on their performance and identifying optimal line-tester combinations. These results coincide with Habiba et al. [41] and Adebayo and Menkir [42], who highlighted the significance of genotype-by-environment interactions in maize breeding. The obtained results emphasized the potential of exploiting genetic variability and combining ability effects to breed hybrids with high yield and resilience to late wilt disease for enhancing sustainable maize production.

Selecting suitable parents for targeted traits is the basis of successful maize breeding and hybrid development. GCA effect estimates effectively identify parents with high genetic potential, enabling the development of offspring with desirable traits through subsequent selection [43]. The inbreds and testers evaluated in the present study exhibited significant variability in GCA effects for most characters, reflecting a wide diversity in their genetic makeup. IL-302 was recognized as an excellent combiner for early flowering among the assessed lines. Moreover, IL-307 and IL-309 were recognized as ideal combiners for breeding short-statured hybrids, which is critical for improving lodging resistance and maximizing yield potential [44]. Additionally, IL-303, IL-304, and IL-306 demonstrated superior positive GCA effects for grain yield, demonstrating their potential as valuable sources of favorable alleles to enhance this key trait. These lines also show promise as effective testers in breeding programs. Moreover, IL-304, IL-306, and IL-307 were identified as excellent combiners for late wilt disease resistance, highlighting their high frequency of favorable alleles for this trait. Notably, IL-304 and IL-306 exhibited positive GCA effects for grain yield and late wilt resistance, making them particularly valuable for breeding disease-resistant and high-yielding hybrids. Among the testers, T1 (SC-167) was identified with notable positive GCA effects for both late wilt resistance and grain yield.

The performance of assessed hybrids exhibited significant differences in their yield-related traits and resistance to late wilt disease. The hybrids IL-304×T1, IL-306×T1, IL-302×T1, IL-301×T1, IL-303×T2, and IL-302×T2 consistently outperformed not only the other hybrids but also the check hybrid (TWC-368) across all measured traits, including number of rows per ear, kernels per row, grain yield, and resistance to late wilt disease. This superior performance indicated their genetic advantage and potential as high-performing hybrids suitable for large-scale cultivation. The ability of these hybrids to combine high productivity with robust disease resistance positions them as valuable contributors to breeding programs focused on enhancing yield stability and resilience in maize production systems.

SCA is crucial in identifying superior hybrids in maize breeding. The findings of this study revealed that crosses IL-301×T1, IL-302×T2, IL-303×T2, and IL-306×T1 were the most effective specific combiners for enhanced grain yield. Among these, IL-303×T2 and IL-306×T1 exhibited high grain yield and favorable SCA effects, highlighting a strong relationship between superior yield performance and SCA effects. This aligns with Ezzat et al. [45] and Bhatla et al. [38], who confirmed similar relationships between positive SCA effects and agronomic performance. Hybrids such as IL-301×T1 and IL-303×T2, which demonstrated significant SCA effects for late wilt disease resistance and grain yield, are promising candidates for further evaluation and development in maize breeding programs. Standard heterosis is another critical metric for selecting superior hybrids by comparing the performance of developed hybrids against commercial cultivars. This measure identifies hybrids with enhanced agronomic traits, increasing their commercial viability and adoption by farmers. The hybrids IL-303×T2 and IL-306×T1 proved significant positive standard heterosis for grain yield and late wilt resistance, outperforming the check hybrid. These results emphasized their potential for providing higher yields and improved disease resistance compared to commercial options. Comparable results were documented in recent research conducted by Sorsa et al. [46] and Habiba et al. [41], documenting significant positive standard heterosis for grain yield, and Kamara et al. [20] highlighted its importance for late wilt resistance. These hybrids, therefore, represent valuable genetic resources for breeding resilient and high-yielding hybrids. Hence, these hybrids have strong potential for commercialization, mainly due to their high yield performance and resistance to late wilt disease. These traits are critical for ensuring stable production under diverse field conditions, especially in regions prone to this pathogen. The hybrids performed well under controlled experimental conditions, but their stability across diverse agroecological zones requires further validation. Genotype-by-environment interaction effects can significantly influence hybrid performance. Multi-environment trials must confirm their adaptability and stability under varying climatic and soil conditions. Furthermore, the hybrid performance depends on agronomic practices, such as irrigation, fertilization, and pest management. Therefore, the response of these hybrids to other abiotic stresses, such as drought and nutrient deficiencies, needs further evaluation to determine the level of management required for optimal performance in farmers’ fields.

The heritability estimates provided valuable insights into the genetic architecture of the evaluated traits. Broad-sense heritability, encompassing additive, dominance, and epistatic genetic variances, was moderately high for most traits. This indicates a substantial genetic contribution to the observed phenotypic variance and underscores the potential for genetic improvement through selection [47,48]. However, the consistently lower narrow-sense heritability values highlight the limited contribution of additive genetic effects to the total genetic variation for many traits. Traits such as ear height, days to silking, and number of rows per ear exhibited relatively high narrow-sense heritability, suggesting a significant additive genetic component. These traits have the potential to respond well to selection, particularly under breeding strategies that focus on additive genetic improvement [49,50]. Their predictability and stability across environments make them attractive targets for genetic enhancement. Conversely, low narrow-sense heritability for other traits suggests a more significant influence of non-additive genetic effects, such as dominance and epistasis, or environmental factors on their expression. For these traits, breeding strategies that exploit heterosis, such as hybrid breeding, may be more effective.

Non-additive genetic effects, such as dominance and epistasis, are crucial in hybrid vigor for essential traits such as yield traits and resistance to late wilt disease [29,51]. For yield traits, dominance effects can improve biomass accumulation, resource-use efficiency, and grain filling. Regarding late wilt resistance, non-additive effects can enhance the hybrid’s ability to activate complex defense pathways. In this study, SCA variance exceeded GCA variances for both late wilt resistance and grain yield, demonstrating that non-additive gene action plays a predominant role in the inheritance of these characters. This observation was corroborated by the dominant genetic variance, which surpassed the additive genetic variance, emphasizing the potential for utilizing heterosis breeding to enhance these traits. These findings align with previous research by Mosa et al. [52], El-Shenawy et al. [53], and Kamara et al. [20], which deduced that non-additive effects primarily govern the inheritance of late wilt disease resistance and grain yield. However, other studies reported contrasting results, where additive genetic effects were identified as the main drivers of trait inheritance. For example, Mosa et al. [54] emphasized the dominance of additive genetic action in expressing late wilt disease resistance, while Mebratu et al. [55] reported similar findings for grain yield. These differences suggest that the relative contributions of additive and non-additive genetic effects may alter depending on the genetic background of the populations studied and the surrounding conditions. Such insights emphasize the importance of altering breeding strategies to the specific genetic architecture of the target traits and the breeding objectives. Understanding the relationships between grain yield, late wilt disease resistance, and other characteristics is essential for improving the efficiency of maize breeding programs. This study demonstrated substantial positive associations between grain yield and key characters such as number of kernels/row, number of rows/ear, plant height, and late wilt disease resistance. These associations highlight the significance of employing these characters as indirect selection criteria to enhance grain yield. Additionally, a positive correlation between grain yield and late wilt disease resistance suggests that improving disease resistance can directly contribute to higher yield potential. These results align with El-Shenawy et al. [53] and Beyene et al. [56], who also disclosed strong positive relationships between disease resistance and grain yield. Such correlations underline the importance of integrating disease resistance into breeding objectives to improve yield and crop resilience. By leveraging these relationships, breeders can adopt a more holistic approach, targeting agronomic performance and disease resistance to develop high-yielding, robust maize hybrids.

Late wilt resistance in maize is governed by a combination of molecular and physiological mechanisms that are associated with agronomic traits [10]. Late wilt resistance involves the activation of pathogen-related genes and the upregulation of secondary metabolites. These compounds reinforce cell walls and inhibit the progression of *M. maydis*. Antifungal proteins and enzymes also play a crucial role in suppressing pathogen growth and spread. Furthermore, resistance is characterized by maintaining vascular integrity, ensuring the continuity of water and nutrient transport despite infection [57]. This is further supported by a robust antioxidant defense system in resistant hybrids, which mitigates oxidative stress induced by pathogens [11,58]. Such mechanisms are critical for sustaining plant health under late wilt pressure. Enhanced resistance assists in reducing pathogen-induced damage, thereby maintaining or even improving yield under disease pressure. However, resistance mechanisms may redirect metabolic resources from growth or reproduction to defense pathways, potentially resulting in yield reductions under non-stress conditions. This trade-off emphasizes the importance of balancing resistance and yield in breeding programs. Hence, further exploration of the genetic basis of these mechanisms, mainly through genomic and transcriptomic approaches, could deepen understanding of the interaction of molecular and physiological mechanisms with agronomic traits. Such insights will provide precise and efficient breeding strategies, enabling the development of hybrids that combine late wilt resistance with optimal yield performance.

## 5. Conclusions

This study revealed significant genetic variability and interaction effects influencing agronomic attributes, grain yield, and late wilt disease resistance among newly developed maize hybrids. The inbreds IL-304 and IL-306 were detected as excellent combiners to enhance grain yield and late wilt disease resistance. Among the assessed hybrids, IL-303×T2, IL-306×T1, and IL-304×T1 consistently demonstrated strong SCA and superior performance across key traits, including grain yield and disease resistance. The predominance of non-additive genetic effects for most characters emphasizes the potential for utilizing heterosis breeding for yield improvement and disease resistance. Moreover, principal component analysis and GGE biplot provided clear insights into optimal combinations, offering strategic guidance for hybrid development. The robust positive associations between grain yield and important characters such as number of rows/ear, number of kernels/row, and late wilt disease resistance emphasized the importance of integrating these characters into selection criteria to improve maize productivity and disease resistance. The insights derived from this study reinforce the value of integrating high-yield potential with robust disease resistance in maize breeding, contributing to developing high-performing and disease-resistant hybrids that address food security challenges.

## Figures and Tables

**Figure 1 life-14-01609-f001:**
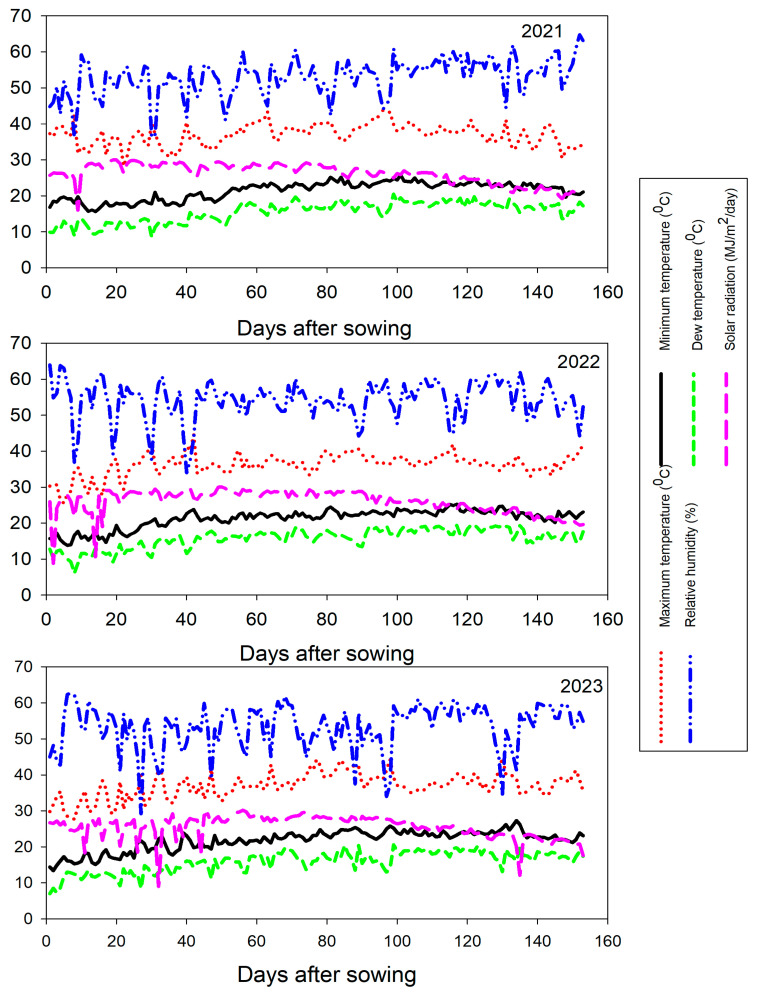
Meteorological characterization for 2021, 2022, and 2023 growing seasons.

**Figure 2 life-14-01609-f002:**
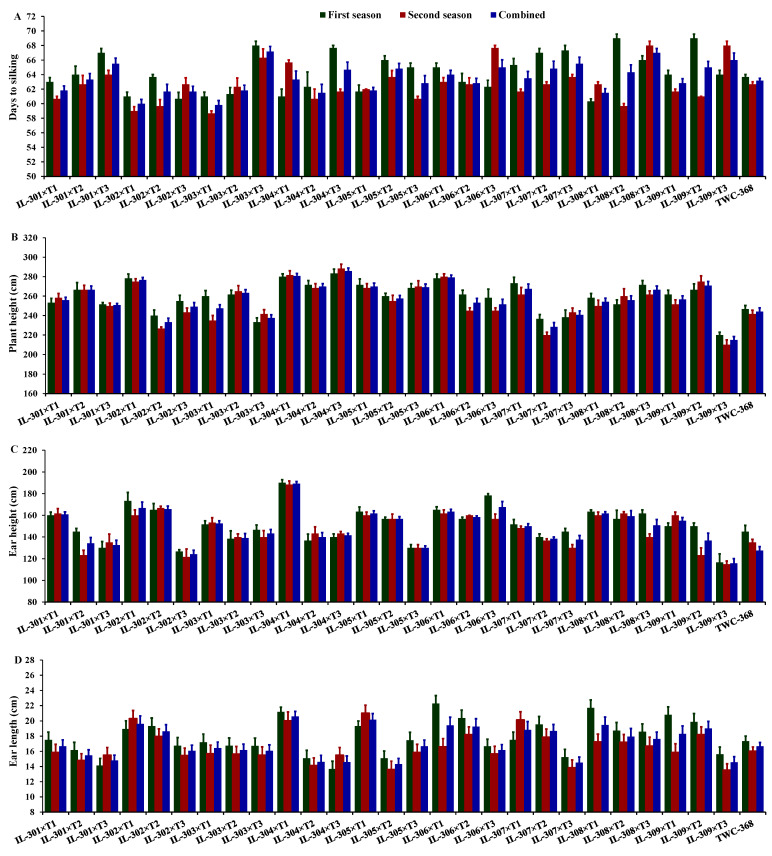
Comparative performance of the assessed twenty-seven hybrids and check hybrid under artificial soil inoculation with *M. maydis*: Days to silking (**A**), plant height (**B**), ear height (**C**), and ear length (**D**). The bars above the columns indicate standard error (SE).

**Figure 3 life-14-01609-f003:**
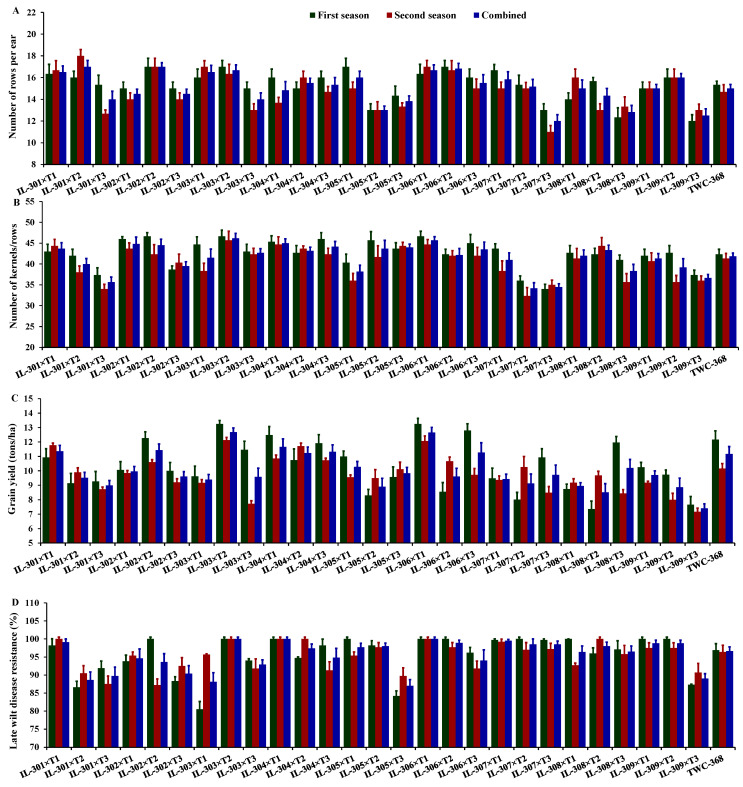
Comparative performance of the assessed twenty-seven hybrids and check hybrid under artificial soil inoculation with *M. maydis*: Number of rows per ear (**A**), number of kernels per row (**B**), grain yield (**C**), and late wilt disease resistance (**D**). The bars above the columns indicate standard error (SE).

**Figure 4 life-14-01609-f004:**
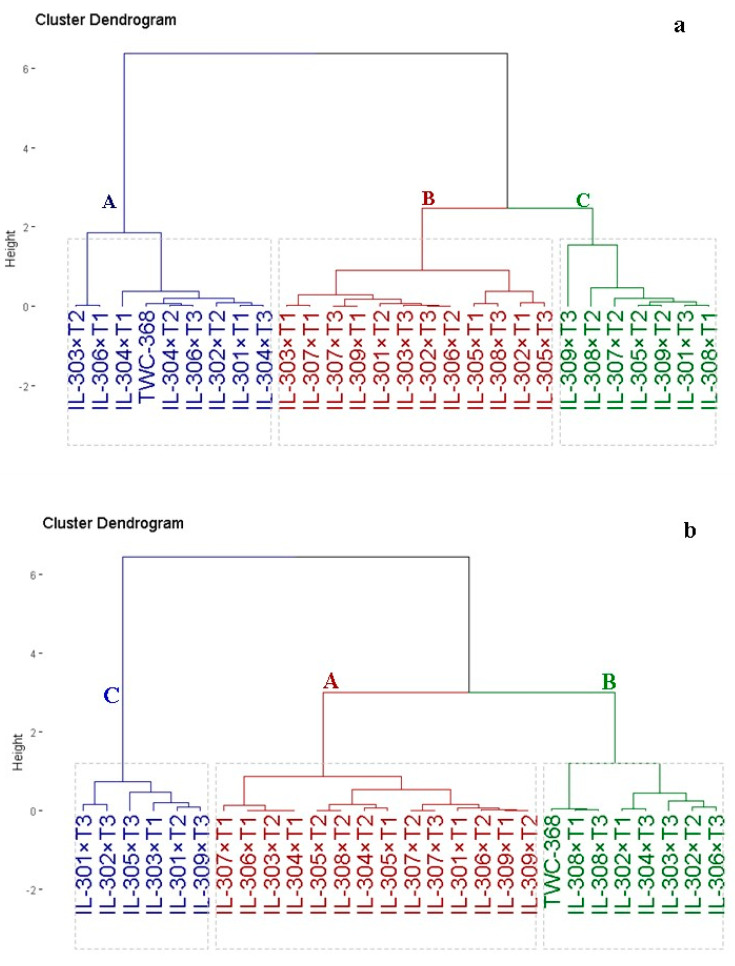
Dendrogram depicting phenotypic distances among 27 hybrids and check hybrids based on yield characters (**a**) late wilt disease resistance (**b**) under artificial soil inoculation with *M. maydis*.

**Figure 5 life-14-01609-f005:**
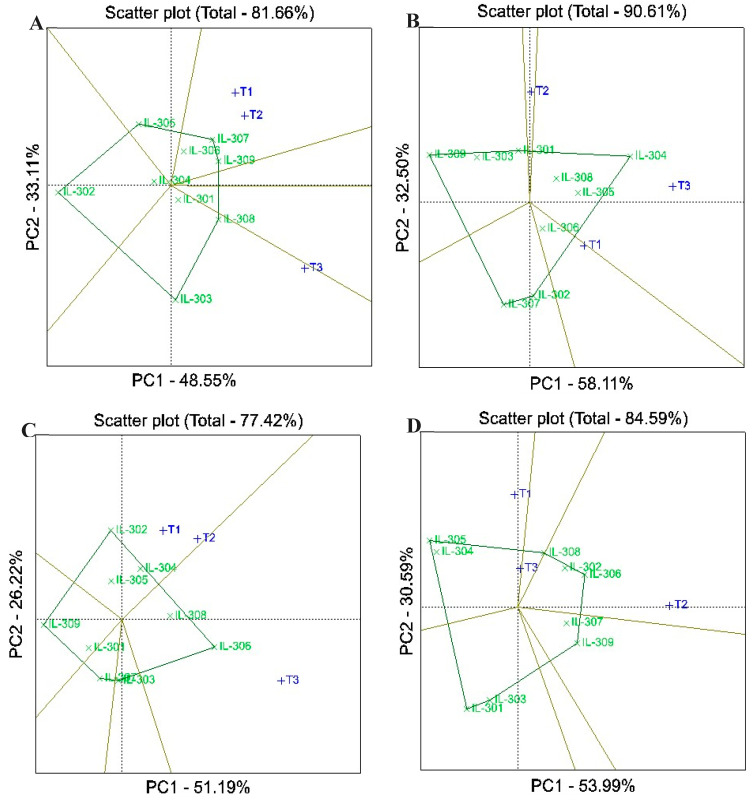
GGE biplot for days to silking (**A**), plant height (**B**), ear height (**C**), and ear length (**D**) of the assessed lines and testers under artificial soil inoculation with *M. maydis*. Lines are presented in green, while testers (T1, T2 and T3) are in blue.

**Figure 6 life-14-01609-f006:**
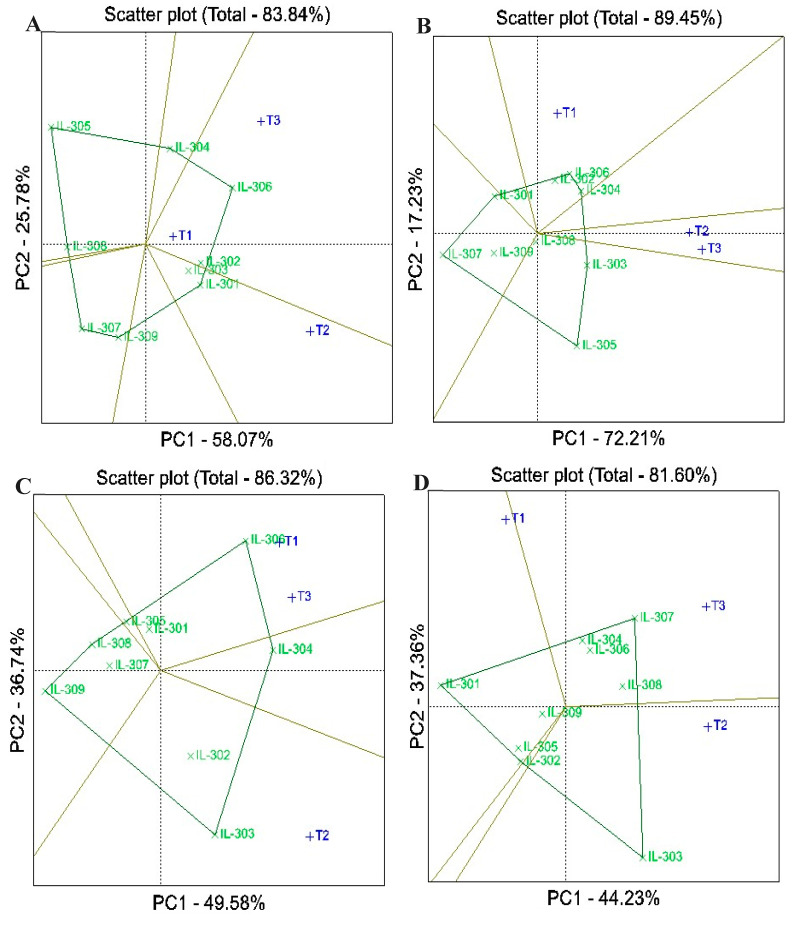
GGE biplot for number of rows per ear (**A**), number of kernels per row (**B**), grain yield (**C**), and late wilt disease resistance (**D**) of the assessed lines and testers under artificial soil inoculation with *M. maydis*. Lines are presented in green, while testers (T1, T2 and T3) are in blue.

**Figure 7 life-14-01609-f007:**
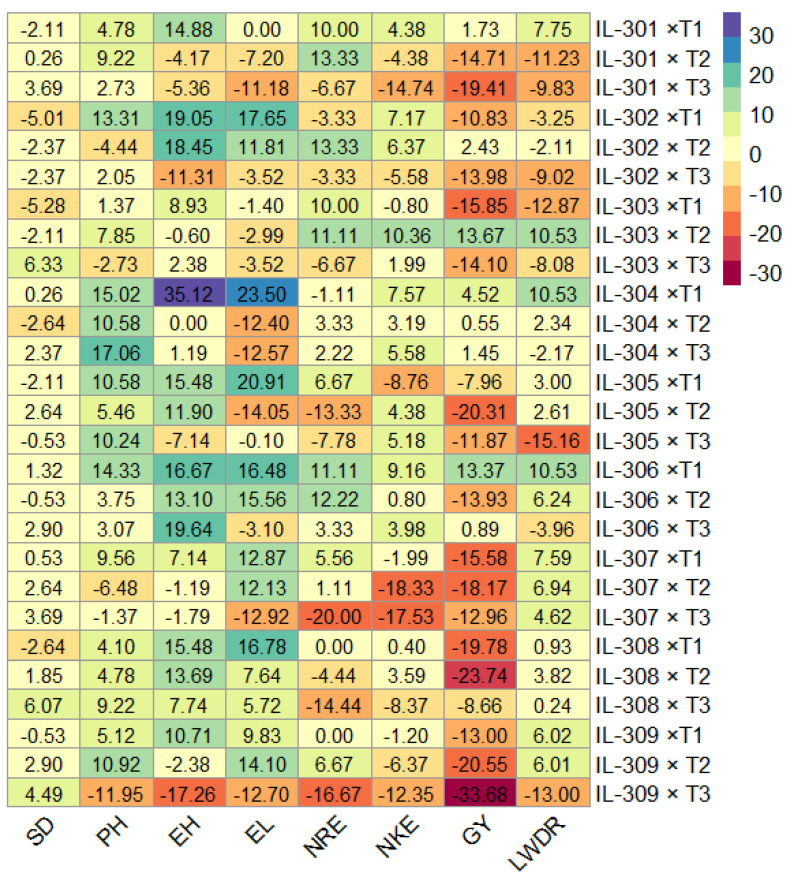
Standard heterosis (%) of maize crosses for agronomic and resistance traits relative to check hybrid TWC-368.

**Figure 8 life-14-01609-f008:**
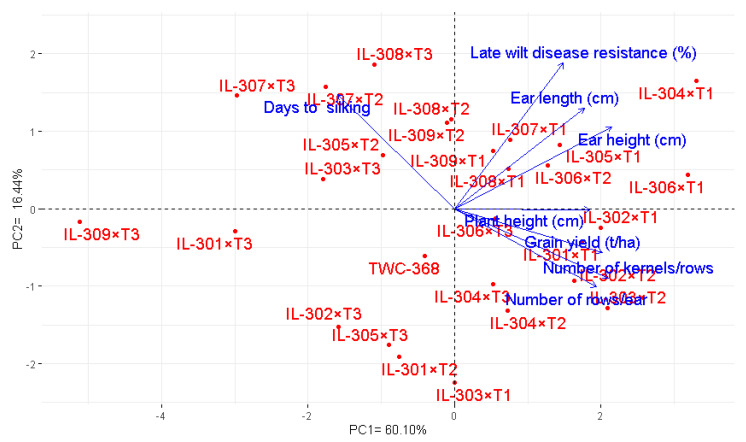
PC-biplot explores the relationship between studied agronomic traits and the assessed maize hybrids under artificial soil inoculation with *M. maydis*.

**Figure 9 life-14-01609-f009:**
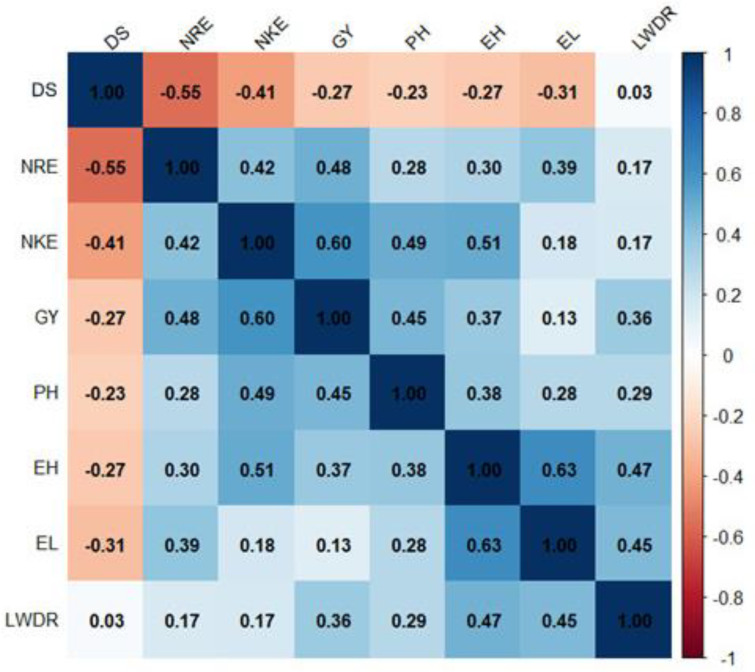
Genotypic correlation coefficients among evaluated traits under artificial soil inoculation with *M. maydis*. DS—days to silking, NRE—number of rows/ear; NKE—number of kernels/row; GY—grain yield; PH—plant height; EH—ear height; EL—ear length; and LWDR—late wilt disease resistance.

**Table 1 life-14-01609-t001:** Mean squares from separate and combined ANOVA for studied agronomic traits.

Source of Variation	df	Days to Silking	Plant Height (cm)	Ear Height (cm)	Ear Length (cm)	Number of Rows/Ear	Number of Kernels /Row	Grain Yield (t/ha)	Late Wilt Disease Resistance (%)
First season									
Crosses	26	21.92 **	727.6 **	819.7 **	16.30 **	6.00 **	34.50 **	8.72 **	212.7 **
Lines	8	17.98 **	812.6 **	691.7 **	16.39 **	6.08 **	50.14 **	9.63 **	149.1 **
Testers	2	66.38 **	1624 **	3194 **	83.08 **	19.27 **	72.09 **	7.74 **	584.6 **
Lines×Testers	16	18.33 **	572.9 **	586.9 **	7.900 *	4.30 **	21.98 **	8.39 **	198.0 **
Error	52	1.85	73.39	70.79	4.31	1.31	7.54	0.98	29.14
Second season									
Crosses	26	20.08 **	1079 **	877.0 **	12.87 **	8.93 **	44.28 **	4.95 **	136.7 **
Lines	8	11.13 **	1210 **	756.6 **	6.14	7.03 **	67.77 *	7.19 **	101.5 *
Testers	2	85.05 **	1052 **	4915 **	53.88 **	45.42 **	34.83 **	14.63 **	463.0 **
Lines×Testers	16	16.42 **	1016 **	432.5 **	11.10 **	5.32 **	33.72 **	2.63 **	113.4 **
Error	52	1.30	83.96	50.84	4.53	1.74	11.47	0.62	43.16
Combined across years									
Years	1	102.7 **	734.7 *	692.75 **	67.66 **	9.39	184.8 *	12.37 *	125.8 *
Replication/Year	4	3.05	133.6	31.33	5.74	5.53 **	35.01 **	2.86 **	23.57
Crosses	26	23.18 **	1688 **	1557 **	23.79 **	12.21 **	69.13 **	9.65 **	250.5 **
Lines	8	21.40 **	1950 **	1393 **	15.91 **	11.75 **	115.9 **	14.86 **	208.0 **
Testers	2	119.3 **	2646 **	8010 **	134.9 **	61.80 **	103.5 **	5.11 **	1044 **
Lines×Testers	16	12.06 **	1437 **	832.6 **	13.85 **	6.24 **	41.48 **	7.61 **	172.5 **
Crosses×Years	26	18.81 **	118.4 *	139.5 **	5.37	2.72 **	9.64	4.03 **	98.87 **
Lines×Years	8	7.71 **	72.57	55.25	6.62	1.36	2.07	1.96 **	42.71
Testers×Years	2	32.17 **	31.02	99.23	2.10	2.89	3.45	17.26 **	3.63
Lines×Testers×Years	26	22.69 **	152.2 **	186.7 **	5.15	3.38 **	14.21	3.41 **	138.9 **
Error	104	1.58	78.67	60.81	4.42	1.52	9.51	0.80	36.15

* and ** denote *p*-values below 0.05 and 0.01 in the same order.

**Table 2 life-14-01609-t002:** GCA estimates for agronomic traits of assessed inbred lines and testers across the first (1st Y) and second (2nd Y) years and combined data (Com.) of both years.

Genotype	Days to Silking			Plant Height (cm)			Ear Height (cm)			Ear Length (cm)		
	1st Y	2nd Y	Com.	1st Y	2nd Y	Com.	1st Y	2nd Y	Com.	1st Y	2nd Y	Com.
Inbred Lines												
IL301	0.38	−0.25	0.07	−2.47	2.90	0.22	−6.42 *	−7.28 **	−6.85 **	−1.92 **	−1.20	−1.56 **
IL302	−2.51 **	−2.25 **	−2.38 **	−1.91	−7.10 *	−4.51 *	3.58	2.16	2.87	0.49	1.31	0.90
IL303	−0.84	−0.25	−0.54	−8.02 **	−8.21 **	−8.12 **	−5.86 *	−2.84	−4.35 *	−0.98	−0.98	−0.98 *
IL304	−0.62	−0.02	−0.32	18.64 **	24.01 **	21.33 **	4.14	11.05 **	7.59 **	−1.21	−0.03	−0.62
IL305	−0.06	−0.58	−0.32	6.98 *	9.01 **	7.99 **	−1.42	1.60	0.09	−0.57	0.24	−0.16
IL306	−0.84	1.75 **	0.46	6.42 *	1.23	3.83	15.25 **	12.16 **	13.70 **	1.91 **	0.23	1.07 *
IL307	2.27 **	−0.02	1.12 **	−10.25 **	−13.77 **	−12.01 **	−5.86 *	−8.95 **	−7.41 **	−0.43	0.69	0.13
IL308	0.83	0.75 *	0.79 **	0.86	1.79	1.33	9.14 **	6.60 **	7.87 **	1.81 **	0.46	1.14 *
IL309	1.38 **	0.86 *	1.12 **	−10.25 **	−9.88 **	−10.06 **	−12.53 **	−14.51 **	−13.52 **	0.90	−0.73	0.08
LSD_(gi) 0.05_	0.89	0.75	0.58	5.60	5.99	4.10	5.50	4.66	3.60	1.36	1.39	0.97
LSD_(gi) 0.01_	1.17	0.98	0.76	7.36	7.87	5.39	7.22	6.12	4.73	1.78	1.83	1.28
Testers												
T1 (SC-167)	−1.80 **	−1.02 **	−1.41 **	8.64 **	6.98 **	7.81 **	11.73 **	14.20 **	12.96 **	1.74 **	1.50 **	1.62 **
T2 (Pioneer-3062)	0.75 **	−1.02 **	−0.14	−2.28	−1.91	−2.10	−1.98	−1.54	−1.76	0.02	−0.19	−0.08
T3 (TWC-360)	1.05 **	2.05 **	1.55 **	−6.36 **	−5.06 **	−5.71 **	−9.75 **	−12.65 **	−11.20 **	−1.76 **	−1.31 **	−1.54 **
LSD_(gi) 0.05_	0.51	0.43	0.33	3.23	3.46	2.37	3.17	2.69	2.08	0.78	0.80	0.56
LSD_(gi) 0.01_	0.67	0.57	0.44	4.25	4.54	3.11	4.17	3.53	2.73	1.03	1.06	0.74
	**Number of rows/ear**			**Number of kernels/row**			**Grain yield** **(t/ha)**			**Late wilt disease resistance (%)**		
	**1st Y**	**2nd Y**	**Com.**	**1st Y**	**2nd Y**	**Com.**	**1st Y**	**2nd Y**	**Com.**	**1st Y**	**2nd Y**	**Com.**
Inbred Lines												
IL301	0.58	0.95 *	0.77 **	−1.72	−1.58	−1.65 *	−0.54	0.36	−0.09	−5.11 **	−2.06	−3.59 *
IL302	0.36	0.17	0.27	1.28	1.75	1.52 *	0.45	0.12	0.29	−1.98	−5.77 **	−3.87 **
IL303	0.69	0.62	0.65 *	2.28 *	1.75	2.02 **	1.12 **	−0.10	0.51 *	−5.72 **	0.12	−2.80 *
IL304	0.36	−0.05	0.15	2.17 *	3.20 **	2.69 **	1.39 **	1.33 **	1.36 **	1.80	4.06	2.93 *
IL305	−0.53	−1.05 *	−0.79 **	0.73	0.31	0.52	−0.70 *	−0.05	−0.38	−2.05	−3.07	−2.56
IL306	1.14 **	1.40 **	1.27 **	2.17 *	2.53 *	2.35 **	1.21 **	1.05 **	1.13 **	4.18 *	2.84	3.51 *
IL307	−0.31	−1.16 **	−0.73 *	−4.60 **	−5.14 **	−4.87 **	−0.84 *	−0.40	−0.62 **	6.45 **	4.00	5.23 **
IL308	−1.31 **	−0.72	−1.01 **	−0.49	0.09	−0.20	−0.98 **	−0.67 *	−0.82 **	1.66	1.11	1.38
IL309	−0.98 *	−0.16	−0.57	−1.83 *	−2.91 **	−2.37 **	−1.11 **	−1.66 **	−1.38 **	0.76	−1.22	−0.23
LSD_(gi) 0.05_	0.75	0.86	0.57	1.79	2.21	1.42	0.65	0.52	0.41	3.53	4.29	2.78
LSD_(gi) 0.01_	0.98	1.13	0.75	2.36	2.91	1.87	0.85	0.68	0.54	4.64	5.64	3.65
Testers												
T1 (SC-167)	0.51 **	0.65 **	0.58 **	1.32 **	0.98	1.15 **	0.32	0.34 **	0.33 **	2.97 **	2.56 **	2.77 **
T2 (Pioneer-3062)	0.47 *	0.84 **	0.65 **	0.51	0.27	0.39	−0.62 **	0.50 **	−0.06	2.40 *	2.21	2.30 **
T3 (TWC-360)	−0.98 **	−1.49 **	−1.23 **	−1.83 **	−1.25	−1.54 **	0.29	−0.84 **	−0.27 *	−5.36 **	−4.78 **	−5.07 **
LSD_(gi) 0.05_	0.43	0.50	0.33	1.04	1.28	0.82	0.37	0.30	0.24	2.04	2.48	1.60
LSD_(gi) 0.01_	0.57	0.65	0.43	1.36	1.68	1.08	0.49	0.39	0.31	2.68	3.26	2.11

* and ** denote *p*-values below 0.05 and 0.01 in the same order.

**Table 3 life-14-01609-t003:** SCA effects of 27 test crosses for days to silking, plant height, ear height, and ear length.

Crosses	Days to Silking			Plant Height (cm)			Ear Height (cm)			Ear Length (cm)		
	1st Y	2nd Y	Com.	1st Y	2nd Y	Com.	1st Y	2nd Y	Com.	1st Y	2nd Y	Com.
IL301×T1	0.14	−0.75	−0.31	−12.53 *	−6.98	−9.75 **	3.27	7.47	5.37	−0.18	−1.02	−0.60
IL301×T2	−1.42	1.25	−0.09	11.73 *	10.25	10.99 **	1.98	−15.12 **	−6.57 *	0.21	−0.40	−0.09
IL301×T3	1.28	−0.49	0.40	0.80	−3.27	−1.23	−5.25	7.65	1.20	−0.04	1.42	0.69
IL302×T1	1.02	−0.42	0.30	11.91 *	19.69 **	15.80 **	6.60	−3.64	1.48	−1.14	0.90	−0.12
IL302×T2	1.14	0.25	0.69	−15.49 **	−19.75 **	−17.62 **	11.98 *	18.77 **	15.37 **	0.96	0.26	0.61
IL302×T3	−2.16 **	0.17	−0.99	3.58	0.06	1.82	−18.58 **	−15.12 **	−16.85 **	0.17	−1.16	−0.49
IL303×T1	−0.64	−2.75 **	−1.70 **	−0.31	−19.20 **	−9.75 **	−5.62	−5.31	−5.46	−1.43	−1.40	−1.41
IL303×T2	−2.86 **	0.91	−0.98	12.28 *	19.69 **	15.99 **	−5.25	−2.90	−4.07	−0.16	0.21	0.02
IL303×T3	3.51 **	1.84 **	2.67 **	−11.98 *	−0.49	−6.23	10.86 *	8.21 *	9.54 **	1.59	1.19	1.39
IL304×T1	−0.86	4.02 **	1.58 **	−6.98	−4.75	−5.86	22.72 **	15.80 **	19.26 **	2.78 *	1.97	2.38 **
IL304×T2	−2.09 **	−0.98	−1.53 **	−4.38	−9.20	−6.79	−16.91 **	−13.46 **	−15.19 **	−1.57	−2.23	−1.90 *
IL304×T3	2.95 **	−3.05 **	−0.05	11.36 *	13.95 **	12.65 **	−5.80	−2.35	−4.07	−1.21	0.26	−0.48
IL305×T1	−0.75	0.91	0.08	−3.64	−3.09	−3.36	1.60	−3.09	−0.74	0.29	2.69 *	1.49
IL305×T2	1.02	2.58 **	1.80 **	−4.38	−7.53	−5.96	8.64	9.32 *	8.98 **	−2.23	−3.03 *	−2.63 **
IL305×T3	−0.27	−3.49 **	−1.88 **	8.02	10.62 *	9.32 *	−10.25 *	−6.23	−8.24 **	1.95	0.34	1.14
IL306×T1	3.36 **	−0.42	1.47 **	3.58	16.36 **	9.97 **	−13.40 **	−11.98 **	−12.69 **	0.76	−1.72	−0.48
IL306×T2	−1.20	−0.75	−0.98	−2.16	−9.75	−5.96	−8.02	2.10	−2.96	0.57	1.57	1.07
IL306×T3	−2.16 **	1.17	−0.49	−1.42	−6.60	−4.01	21.42 **	9.88 *	15.65 **	−1.33	0.15	−0.59
IL307×T1	0.58	0.02	0.30	15.25 **	13.02 *	14.14 **	−5.62	−4.20	−4.91	−1.66	1.37	−0.15
IL307×T2	−0.31	1.02	0.36	−10.49 *	−19.75 **	−15.12 **	−3.58	−0.12	−1.85	2.10	0.77	1.43
IL307×T3	−0.27	−1.05	−0.66	−4.75	6.73	0.99	9.20	4.32	6.76 *	−0.43	−2.15	−1.29
IL308×T1	−2.98 **	0.25	−1.36 **	−10.86 *	−14.20 **	−12.53 **	−8.95	−8.09 *	−8.52 **	0.29	−1.29	−0.50
IL308×T2	3.14 **	−2.75 **	0.19	−6.60	4.69	−0.96	−1.91	9.32 *	3.70	−0.97	0.33	−0.32
IL308×T3	−0.16	2.51 **	1.17 *	17.47 **	9.51	13.49 **	10.86 *	−1.23	4.81	0.68	0.96	0.82
IL309×T1	0.14	−0.86	−0.36	3.58	−0.86	1.36	−0.62	13.02 **	6.20	0.28	−1.50	−0.61
IL309×T2	2.58 **	−1.53 *	0.52	19.51 **	31.36 **	25.43 **	13.09 **	−7.90	2.59	1.10	2.52 *	1.81 *
IL309×T3	−2.72 **	2.40 **	−0.16	−23.09 **	−30.49 **	−26.79 **	−12.47 *	−5.12	−8.80 **	−1.38	−1.02	−1.20
LSD_0.05_	1.54	1.29	1.00	9.69	10.37	7.10	9.52	8.07	6.24	2.35	2.41	1.68
LSD_0.01_	2.02	1.70	1.32	12.74	13.63	9.33	12.51	10.60	8.20	3.09	3.17	2.21

* and ** denote *p*-values below 0.05 and 0.01 in the same order.

**Table 4 life-14-01609-t004:** SCA effects of 27 test crosses for number of rows/ear, number of kernels/row, grain yield, and late wilt disease resistance.

Crosses	Number of Rows/Ear			Number of Kernels/Row			Grain Yield (t/ha)			Late Wilt Disease Resistance (%)		
	1st Y	2nd Y	Com.	1st Y	2nd Y	Com.	1st Y	2nd Y	Com.	1st Y	2nd Y	Com.
IL301×T1	−0.06	0.23	0.09	0.90	4.58 *	2.74 *	0.83	1.30 **	1.07 **	5.33	8.98 *	7.16 **
IL301×T2	−0.36	1.38	0.51	0.72	−1.05	−0.17	−0.02	−0.74	−0.38	−7.30 *	−8.38 *	−7.84 **
IL301×T3	0.42	−1.62 *	−0.60	−1.62	−3.53	−2.57 *	−0.81	−0.56	−0.69	1.96	−0.61	0.68
IL302×T1	−1.17	−1.65 *	−1.41 **	0.90	0.58	0.74	−1.04	−0.38	−0.71	−3.45	0.43	−1.51
IL302×T2	0.86	1.16	1.01 *	2.38	−0.05	1.17	2.11 **	0.21	1.16 **	7.30 *	−7.54 *	−0.12
IL302×T3	0.31	0.49	0.40	−3.28 *	−0.53	−1.91	−1.07	0.17	−0.45	−3.85	7.11	1.63
IL303×T1	−0.51	0.90	0.20	−1.43	−4.75 *	−3.09 *	−2.15 **	−0.84	−1.50 **	−15.71 **	−5.12	−10.41 **
IL303×T2	0.53	0.05	0.29	1.38	3.28	2.33	2.42 **	1.96 **	2.19 **	11.04 **	7.15	9.10 **
IL303×T3	−0.02	−0.95	−0.49	0.05	1.47	0.76	−0.28	−1.11 *	−0.69	4.67	−2.03	1.32
IL304×T1	−0.17	−1.77 *	−0.97	−0.65	0.14	−0.26	0.44	−0.58	−0.07	2.96	2.86	2.91
IL304×T2	−1.14	0.38	−0.38	−2.51	−0.16	−1.33	−0.35	0.11	−0.12	−9.82 **	3.21	−3.30
IL304×T3	1.31 *	1.38	1.35 **	3.16 *	0.02	1.59	−0.09	0.47	0.19	6.86 *	−6.06	0.40
IL305×T1	1.72 **	0.57	1.14 *	−4.21 **	−5.64 **	−4.93 **	1.05	−0.51	0.27	6.80 *	−2.26	2.27
IL305×T2	−2.25 **	−1.62 *	−1.93 **	1.94	0.73	1.33	−0.71	−0.73	−0.72 *	1.45	3.37	2.41
IL305×T3	0.53	1.05	0.79	2.27	4.91 *	3.59 **	−0.35	1.23 **	0.44	−8.25 **	−1.11	−4.68
IL306×T1	−0.62	0.12	−0.25	0.68	0.80	0.74	1.40 *	0.90 *	1.15 **	0.58	4.08	2.33
IL306×T2	0.09	−0.40	−0.15	−2.84	−1.16	−2.00	−2.37 **	−0.66	−1.51 **	1.15	−2.54	−0.70
IL306×T3	0.53	0.27	0.40	2.16	0.36	1.26	0.97	−0.25	0.36	−1.73	−1.54	−1.63
IL307×T1	1.16	0.68	0.92	4.46 **	2.14	3.30 **	−0.32	−0.35	−0.33	−3.60	0.04	−1.78
IL307×T2	−0.14	0.49	0.18	−2.40	−3.16	−2.78 *	−0.85	0.39	−0.23	−1.13	−2.57	−1.85
IL307×T3	−1.02	−1.17	−1.10 *	−2.06	1.02	−0.52	1.17 *	−0.04	0.57	4.73	2.53	3.63
IL308×T1	−0.51	1.23	0.36	−0.65	−0.09	−0.37	−0.94	−0.26	−0.60	3.09	−9.81 **	−3.36
IL308×T2	1.20	−1.95 *	−0.38	−0.17	3.62	1.72	−1.38 *	0.08	−0.65	−7.26 *	6.17	−0.55
IL308×T3	−0.69	0.72	0.01	0.83	−3.53	−1.35	2.31 **	0.19	1.25 **	4.17	3.65	3.91
IL309×T1	0.16	−0.32	−0.08	0.01	2.25	1.13	0.72	0.72	0.72 *	4.00	0.80	2.40
IL309×T2	1.20	0.49	0.85	1.49	−2.05	−0.28	1.15 *	−0.62	0.26	4.57	1.14	2.85
IL309×T3	−1.36 *	−0.17	−0.77	−1.51	−0.20	−0.85	−1.86 **	−0.11	−0.98 **	−8.56 **	−1.94	−5.25 *
LSD_0.05_	1.30	1.49	0.99	3.11	3.83	2.47	1.12	0.89	0.72	6.11	7.43	4.81
LSD_0.01_	1.70	1.96	1.30	4.08	5.04	3.24	1.47	1.18	0.94	8.03	9.77	6.32

* and ** denote *p*-values below 0.05 and 0.01 in the same order.

**Table 5 life-14-01609-t005:** Genetic parameter estimates for studied agronomic traits across two years.

Trait	σ^2^ GCA	σ^2^ SCA	σ^2^ GCA /σ^2^ SCA	Additive Variance (σ^2^ A)	Dominance Variance (σ^2^ D)	h^2^b	h^2^n
Days to silking	1.91	1.75	1.09	3.82	1.75	69.9	36.51
Plant height (cm)	61.65	226.4	0.27	123.3	226.36	78.54	16.81
Ear height (cm)	128.9	128.6	1.0	257.8	128.64	80.9	40.49
Ear length (cm)	1.97	1.57	1.25	3.94	1.57	44.5	24.76
Number of rows/ear	0.98	0.79	1.24	1.96	0.79	53.66	29.77
Number of kernels/row	2.78	5.33	0.52	5.56	5.33	46.04	15.79
Grain yield (tons/ha)	0.26	1.13	0.23	0.52	1.13	63.39	11.64
Late wilt disease resistance (%)	16.38	22.73	0.72	32.76	22.73	51.97	21.77

## Data Availability

The data presented in this study are available upon request from the corresponding author.

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
