# Peer review of "Unraveling Genetic Variation and Inheritance Patterns in Newly Developed Maize Hybrids for Improving Late Wilt Disease Resistance and Agronomic Performance Under Artificial Inoculation Conditions"

_life, 2024, doi:10.3390/life14121609_

Round 1
Reviewer 1 Report
Comments and Suggestions for Authors
· How were the GGE biplots and PCA validated? Were cross-validation or other statistical methods used to ensure robustness?
· Were the two growing seasons under comparable environmental conditions, or were significant seasonal differences accounted for in the analysis?
· How do IL-306, IL-304, and IL-303 perform under non-inoculated conditions? Is there a trade-off between yield and disease resistance?
· Provide a comparison of these hybrids' performance with existing commercial maize hybrids. This would contextualize the significance of the findings.
· Discuss potential molecular or physiological mechanisms that contribute to late wilt resistance and their association with agronomic traits.
· Did the study consider genotype-by-environment interactions for the assessed traits? How stable are the hybrids' performances across environments?
· Highlight how non-additive genetic effects influence hybrid vigor for grain yield and late wilt resistance. Consider integrating quantitative genetic models for deeper insights.
· Expand the discussion on heritability estimates. How do these align with previous studies, and what are the implications for future breeding programs?
· Discuss the feasibility of scaling up these hybrids for large-scale cultivation. Are there any limitations to their use under field conditions?
· Have any hybrids been tested under natural field conditions where the disease is prevalent, without artificial inoculation?
· Has the economic feasibility of deploying these resistant hybrids been evaluated? For example, how do their input costs compare to their expected benefits in mitigating yield losses?
Author Response
Responses to Reviewers' Comments
Reviewer 1:
Re: We would like to thank the Reviewer for the time he dedicated to our manuscript.
How were the GGE biplots and PCA validated? Were cross-validation or other statistical methods used to ensure robustness?
Re: We appreciate the reviewer question regarding the validation of GGE biplots and PCA analysis. In this study, the PCA was validated through genotypic correlation analysis, where we examined the correlation among evaluated traits to ensure that the relationships identified by PCA were consistent with the underlying trait associations observed in the dataset. This approach reinforces the robustness of PCA in representing the data structure and associations among traits. The GGE biplot analysis results were cross-validated using line×tester analysis of Kempthorne (1957). The line×tester analysis provided independent confirmation of the suitability of line×tester combinations. By aligning the findings from both approaches, we demonstrated that GGE biplots effectively identified superior hybrids and optimal parental combinations, thus supporting their reliability and applicability in our breeding program. Additionally, the consistency of our results with previously established patterns in agronomic studies provides further validation.
Were the two growing seasons under comparable environmental conditions, or were significant seasonal differences accounted for in the analysis?
Re: We appreciate the reviewer inquiry regarding the environmental comparability of the growing seasons. While the two seasons were conducted under consistent management practices and agronomic protocols, some environmental variability was observed. To account for these variations, we performed both a combined analysis and separate analyses for each season. This approach allowed to evaluate the performance of hybrids across seasons while accounting for seasonal differences in the environmental conditions.
How do IL-306, IL-304, and IL-303 perform under non-inoculated conditions? Is there a trade-off between yield and disease resistance?
Re: Late wilt disease poses a critical threat to maize cultivation in Egypt, making it essential to evaluate all inbred lines under artificial inoculation across multiple growing seasons. This approach allows for a thorough assessment of their disease resistance and yield potential under high disease pressure. The performance of IL-306, IL-304, and IL-303 under non-inoculated conditions was not the primary focus of this study. However, we recognize the importance of such evaluations in understanding potential trade-offs between yield and disease resistance. To address this, our future research plans will include trials under non-inoculated conditions to gain deeper insights into these dynamics and further inform breeding strategies. Thank you for the valuable suggestion.
Provide a comparison of these hybrids' performance with existing commercial maize hybrids. This would contextualize the significance of the findings.
Re: The agronomic performance of the developed hybrids was compared with the high-yielding commercial hybrid TWC-368 to provide a clear context for the significance of the findings. Additional clarifications and comparisons have been incorporated into the discussion (lines 522-531).
Discuss potential molecular or physiological mechanisms that contribute to late wilt resistance and their association with agronomic traits.
Re: Thank you for suggesting this important point. The Discussion section has incorporated more information to address potential molecular and physiological mechanisms contributing to late wilt resistance (lines 614-632).
Did the study consider genotype-by-environment interactions for the assessed traits? How stable are the hybrids' performances across environments?
Re: The study accounted for genotype-by-year (G×Y) interactions to evaluate the stability of the developed hybrids across growing seasons. G×Y interaction was analyzed using a combined analysis of variance with the year included as factor to assess its impact on the traits studied. The results of this study identified promising high-yielding hybrids and resistant to late wilt disease, consistently demonstrating superior performance across two growing seasons. Future trials are planned across multiple environments and locations further to evaluate the stability and adaptability of these promising hybrids. These multi-environment trials will provide a more comprehensive understanding of the hybrids performances under diverse environmental conditions, ensuring their suitability for broader cultivation or specific agroecological zones.
Highlight how non-additive genetic effects influence hybrid vigor for grain yield and late wilt resistance. Consider integrating quantitative genetic models for deeper insights.
Re: A paragraph has been added to the discussion section (lines 580-599) emphasizing the influence of non-additive genetic effects on grain yield and late wilt resistance. This discussion integrates quantitative genetic models, such as Specific Combining Ability (SCA) and heterosis models, to provide deeper insights into the role of non-additive effects. SCA captures the dominance and epistatic interactions between parental lines, which are critical for hybrid vigor. Additionally, heterosis models illustrate how hybrids outperform their parents due to these non-additive interactions, particularly for traits like yield and disease resistance. This inclusion highlights the significant contribution of non-additive genetic effects to the superior performance of hybrids in the study.
Expand the discussion on heritability estimates. How do these align with previous studies, and what are the implications for future breeding programs?
Re: The discussion on heritability estimates has been expanded to provide a more comprehensive understanding of their alignment with their implications for future breeding programs (lines 564-579).
Discuss the feasibility of scaling up these hybrids for large-scale cultivation. Are there any limitations to their use under field conditions?
Re: The feasibility of scaling up these hybrids for large-scale cultivation has been discussed as suggested (lines 552-563).
Have any hybrids been tested under natural field conditions where the disease is prevalent
Re: The current study was conducted under artificial inoculation conditions, allowing for a controlled and uniform evaluation of hybrid resistance to late wilt disease. This method ensured exposure to consistent pathogen levels, enabling accurate assessment of the hybrid resistance potential. However, the promising hybrids identified through this study will advance to the next testing phase, which includes multi-environmental trials. These trials will be conducted in farmer field conditions, exposing the hybrids to natural disease prevalence and diverse environmental factors. This step is essential for a more comprehensive understanding of their field performance, adaptability, and resilience.
Clarifications have been added to the discussion section (lines 552–563).
Has the economic feasibility of deploying these resistant hybrids been evaluated? For example, how do their input costs compare to their expected benefits in mitigating yield losses?.
Re: The current study focused on identifying and evaluating resistant hybrids under controlled conditions and has not assessed the economic feasibility of deploying the resistant hybrids. However, considering the financial aspects, such as input costs relative to their expected benefits in mitigating yield losses, is crucial for determining their practical adoption by farmers. Future research phases, particularly during the multi-environmental trials, will provide an opportunity to gather data on the hybrid performance under actual farming conditions. Integrating economic feasibility studies into future research will ensure that these hybrids are agronomically viable and economically attractive for large-scale deployment.
We greatly appreciate the reviewer valuable feedback and insightful comments, which have significantly contributed to strengthening the manuscript.
Reviewer 2 Report
Comments and Suggestions for Authors
The topics presented in the paper are important contributions to the development of progress in crop production, including maize in the context of ongoing climate change. I agree with the authors that climate change is causing new challenges in crop production and the need to look for new varieties and pursue the development of hybrids. The development of fungal diseases in maize is becoming more frequent and causes additional costs in cultivation. It is therefore extremely important to search for their hybrids to cope with this progressive negative situation, especially in areas with dry climate. The topic of the study is important and deserve appreciation.
However, the structure of the study needs some adjustment. After a well-prepared introduction, a subsection with a literature review is missing,. In this chapter it would be worthwhile to broadly describe the impact of progressive climate change on the development of fungal diseases of crops in agriculture on a global basis. Such an additional subsection would strengthen the international aspect and make the study even more interesting. I congratulate the research method used, you can see that the authors use the method well and freely, which makes the results even more interesting. I am glad that the results of the study are prepared on solid figures/charts and tables. Such coverage is important for readers to analyze the results. The discussion of the results has been prepared very well and does not need to be revised. The conclusions are well established.
The study needs to be expanded in terms of the addition of a subsection with a review of the international literature in terms of reviewing the results on the relationship between climate change and the development of fungal diseases in agricultural crops on a global basis.
The study is very interesting and should be published after this correction. Congratulations on an important and timely topic.
Author Response
Reviewer 2:
The topics presented in the paper are important contributions to the development of progress in crop production, including maize in the context of ongoing climate change. I agree with the authors that climate change is causing new challenges in crop production and the need to look for new varieties and pursue the development of hybrids. The development of fungal diseases in maize is becoming more frequent and causes additional costs in cultivation. It is therefore extremely important to search for their hybrids to cope with this progressive negative situation, especially in areas with dry climate. The topic of the study is important and deserve appreciation.
Re: We sincerely thank the reviewer for thoughtful and encouraging feedback. The significance of addressing fungal diseases and the value of our methodology and results is deeply appreciated.
However, the structure of the study needs some adjustment. After a well-prepared introduction, a subsection with a literature review is missing. In this chapter it would be worthwhile to broadly describe the impact of progressive climate change on the development of fungal diseases of crops in agriculture on a global basis. Such an additional subsection would strengthen the international aspect and make the study even more interesting.
Re: We greatly appreciate your suggestion to enhance the introduction by including a subsection on the impact of climate change on the development of fungal diseases in crops, focusing on a global perspective. This is a valuable addition, and we agree that it would strengthen the international relevance of the study and provide readers with a broader context for our work. We have addressed this suggestion in lines 70–81.
I congratulate the research method used, you can see that the authors use the method well and freely, which makes the results even more interesting. I am glad that the results of the study are prepared on solid figures/charts and tables. Such coverage is important for readers to analyze the results. The discussion of the results has been prepared very well and does not need to be revised. The conclusions are well established.
Re: We would like to thank the reviewer for positive assessment and encouraging feedback.
The study needs to be expanded in terms of the addition of a subsection with a review of the international literature in terms of reviewing the results on the relationship between climate change and the development of fungal diseases in agricultural crops on a global basis.
Re: The relationship between climate change and the development of fungal diseases in agricultural crops has been clarified in the discussion as suggested (lines 475-483).
The study is very interesting and should be published after this correction. Congratulations on an important and timely topic.
Re: We would like to thank the reviewer for the positive assessment and encouraging feedback. Thank you once again for recognizing the significance of our work.